# A Deep Transfer Learning-Based Visual Inspection System for Assembly Defects in Similar Types of Manual Tool Products

**DOI:** 10.3390/s25061645

**Published:** 2025-03-07

**Authors:** Hong-Dar Lin, Hsiang-Ling Wu, Chou-Hsien Lin

**Affiliations:** 1Department of Industrial Engineering and Management, Chaoyang University of Technology, Taichung 413310, Taiwan; s11115606@gm.cyut.edu.tw; 2Department of Civil, Architectural, and Environmental Engineering, The University of Texas at Austin, Austin, TX 78712-0273, USA; chslin@utexas.edu

**Keywords:** manual tool assembly, automated visual inspection, deep learning, transfer learning

## Abstract

This study introduces an advanced inspection system for manual tool assembly, focusing on defect detection and classification in flex-head ratchet wrenches as a modern alternative to traditional inspection methods. Using a deep learning R-CNN approach with transfer learning, specifically utilizing the AlexNet architecture, the system accurately identifies and classifies assembly defects across similar tools. This study demonstrates how a pre-trained defect detection model for older manual tool models can be efficiently adapted to new models with only moderate amounts of new samples and fine-tuning. Experimental evaluations at three assembly stations show that the AlexNet model achieves a classification accuracy of 98.67% at the station with the highest defect variety, outperforming the R-CNN model with randomly initialized weights. Even with a 40% reduction in sample size for new products, the AlexNet model maintains a classification accuracy of 98.66%. Additionally, compared to R-CNN, it improves average effectiveness by 9% and efficiency by 26% across all stations. A sensitivity analysis further reveals that the proposed method reduces training samples by 50% at 50% similarity while enhancing effectiveness by 13.06% and efficiency by 5.31%.

## 1. Introduction

Products are typically assembled using screws to connect various parts or modules, making manual tools essential across industries. As demand for customization grows, assembly and disassembly processes have become more complex, posing challenges for efficiency and quality management. To address these issues, industries have adopted automated machines and industrial robots, which enhance assembly efficiency, reduce labor costs, improve product quality, and increase production line flexibility [1,2].

Product assembly quality directly impacts the final product’s functionality and safety, with severe defects potentially leading to user harm or property damage. This study focuses on the assembly process of wrenches, a key manual tool, to develop an automated defect detection system for precision assembly. Despite well-designed components, improper assembly can render a wrench unusable. In many factories, assembly and inspection are performed manually, relying on visual inspection, which often fails to detect defects. Missed defects can lead to product returns and customer dissatisfaction. To address these challenges, this study uses flex-head ratchet wrenches as the experimental subject to examine and exploit an assembly defect inspection system tailored for manual tools requiring precision assembly. Furthermore, this application is extended by utilizing transfer learning to develop defect inspection models for other types of manual tools, requiring only a small number of new images. This approach aims to better meet the practical demands of the industry.

A flex-head ratchet wrench consists of 17 components (15 types). Figure 1 illustrates the assembly structure of a 1/4″ DR 72T flex-head ratchet wrench, including (a) three views (front, back, side) of the assembled product, (b) a parts diagram, and (c) an exploded view. Figure 2 lists the assembly parts. The assembly process is divided into three stations: Station 1: The ratchet head, pick, left pawl, right pawl, and springs are assembled. Good workpieces are sent to the second station via a conveyor belt. Station 2: The torque head, dust cover, T8 screws, and washer are installed. Assembled workpieces are transported to the third station. Station 3: the T7 screw, handle, button, T15 screw, spring, and steel ball are installed, completing the assembly. Figure 3 outlines the assembly process and parts used at each station.

This study examines three common assembly anomalies: missing components, misplaced components, and extra components, which frequently occur during flex-head ratchet wrench assembly. Due to their similar appearance, these anomalies are prone to misjudgment, even with manual inspection. Each anomaly kind can be subdivided into various defect types based on the operations at each assembly station, totaling nearly 30 defect types. Historical records indicate that about 35% of products on the production line have defects, with missing parts being the most frequent (60%), followed by misplaced parts (30%), extra parts (5%), and others (5%). Inspection begins by identifying the anomaly kind and then pinpointing the specific defect type. This classification is crucial for process improvement, enabling engineers to trace root causes and implement corrective measures. Figure 4 outlines the relationship between anomaly kinds and their corresponding defect types.

When the assembly line for manual tools switches to producing a different product model (line change), the detection system can apply transfer learning to adapt the trained detection model for the previous product to the new model. By moderately modifying and fine-tuning the existing model, it can be transferred to suit the detection needs of the new product model, requiring only a limited amount of new samples instead of relying on a large dataset for retraining. The transfer learning method extends this assembly defect detection system for application and development. It targets images with characteristics similar to ratchet wrench parts, such as different models of wrenches, other types of manual tools, or images of industrial components. The transfer learning method is used to migrate the pre-trained defect detection model to the recognition model of new product models and to conduct tests and verification to promote the practical applicability of this automated assembly anomaly and defect detection system for industrial components. New product models only need a small number of samples to complete the training and testing of the model. They can achieve good defect detection results, which can shorten the training time and reduce the training cost, making them suitable for applications with less data [3].

Since most product data within the same industry shares some degree of correlation, transfer learning can be applied to adapt existing models into new ones, thereby accelerating and optimizing the learning efficiency of the new models. This approach eliminates the need for models to relearn entirely from scratch, as is typical with most network models. This method is particularly beneficial for assembly processes involving diverse product types and frequent production line changes. This study utilizes the pre-trained defect detection model for ratchet wrenches from the previous study [4] as the source domain. Using transfer learning, the model is adapted to the target domain for defect identification in flex-head ratchet wrenches, followed by testing and validation. Table 1 provides a detailed comparison of the number of product components and defect-related items between these two domains.

Traditional automation systems excel in reducing costs and improving efficiency but lack the flexibility and adaptability of human inspectors, who can identify subtle visual and functional defects and interpret changes in workpiece appearance that impact quality. Deep learning, with its ability to mimic human intelligence using neural networks with thousands of layers, enables automated visual inspection systems to surpass human and traditional machine vision processes [2]. These systems combine human adaptability with the speed and precision of machines, effectively identifying anomalies and tolerating natural variations in complex patterns. This study applies deep learning models combined with transfer learning techniques to the assembly inspection of various similar manual tool products, focusing on defect localization and classification. The challenge lies in achieving accurate and efficient defect detection due to the numerous potential defect types and their subtle visual differences. Additionally, another contribution of this study is demonstrating how a pre-trained detection model for older manual tool models can be adapted and transferred to suit new manual tool models with only a moderate number of new samples and fine-tuning.

This study proposes using the R-CNN model combined with AlexNet and transfer learning for assembly defect detection in various similar manual tool products. This defect detection approach offers robust feature extraction capabilities and high detection accuracy. Utilizing AlexNet to extract high-level features from images effectively addresses subtle anomalies such as surface defects. The region proposal mechanism of R-CNN is well-suited for precise defect localization, particularly for small component misplacements, missing parts, and other minor flaws. Additionally, AlexNet’s pre-trained weights, integrated with transfer learning, allow the model to quickly adapt to new defect detection scenarios with minimal sample requirements for fine-tuning. Although R-CNN’s region-by-region detection method is computationally intensive, its exceptional accuracy and flexibility make it advantageous for tasks requiring detailed and precise defect detection.

This article is structured as follows: first, a review of current automated visual inspection methods for industrial assembly defects; next, a presentation of the proposed deep learning models with transfer learning for detecting and localizing defects in similar manual tools; followed by tests evaluating the models’ performance against traditional methods; and finally, a summary of contributions and future research directions.

## 2. Related Works

In recent years, machine vision has gradually replaced traditional manual visual inspection methods. Currently, defect inspection can be broadly categorized into surface defect detection and assembly defect inspection. Most existing studies focus on surface defects, which primarily affect product appearance. However, assembly defects impact the structural integrity of the workpiece, potentially rendering the product non-functional. Current research on assembly defect inspection typically addresses only one or two kinds of assembly anomalies. In contrast, this study examines three kinds of assembly anomalies and extends the inspection capability to similar types of products. This approach helps prevent subsequent assembly defects caused by defects in earlier assembly stages.

### 2.1. Assembly Defect Inspection Methods

Chen et al. [5] developed a two-stage deep learning framework for train component defect detection, addressing displacement, breakage, and absence with effective results. Xie et al. [6] used genetic programming to detect PCB defects, including missing, misplaced, and incorrect components. Tan et al. [7] applied a neural network to identify misplaced and missing areas in images, achieving strong performance by modeling defects as puzzle-like scenarios. Zhang et al. [8] proposed a residual network with knowledge encoding for chainsaw assembly defect detection, targeting blade breakage, part inversion, and missing components with high accuracy in real-time scenarios. Liu et al. [9] employed Mask R-CNN for chassis assembly defect detection, effectively addressing issues caused by missing or incorrect assemblies. Hao et al. [10] used Faster R-CNN to identify foreign objects and component defects on transmission lines, demonstrating robust performance in safeguarding grid operations.

### 2.2. Deep Learning-Based Object Detection Approaches

Convolutional neural networks (CNNs) are widely used in deep learning due to their strong pattern recognition and ability to process image, audio, and signal data. R-CNN, introduced by Girshick et al. [11] in 2014, was the first algorithm to integrate deep learning with object detection, achieving a 30% mAP improvement in VOC 2012 and establishing a key milestone. Cui et al. [12] combined CenterNet with R-CNN for traffic target detection, using object center localization instead of traditional RPN for region extraction, balancing accuracy and speed. Sun et al. [13] improved Faster R-CNN for wheel hub defect detection, achieving simpler, faster, and more accurate results. Fang et al. [14] developed an attention-based R-CNN for detecting defects in complex industrial settings, addressing issues like diverse defect types and unclear boundaries with high accuracy and robustness. Wang et al. [15] enhanced Cascade R-CNN for metal defect detection, improving image feature extraction with an advanced backbone network and contrast histogram equalization, yielding effective results.

In 2012, Krizhevsky et al. [16] introduced AlexNet, the first deep convolutional neural network, during the ImageNet competition, revolutionizing deep learning by significantly improving image classification accuracy. Boudiaf et al. [17] applied AlexNet with transfer learning to detect six surface defect types in hot-rolled steel strips, demonstrating its effectiveness. Agarwal et al. [18] used AlexNet in a CNN framework to classify lung CT images for cancer detection, achieving higher accuracy than other methods. Chen et al. [19] developed a 3D AlexNet for automatic prostate cancer segmentation in MRI images, outperforming ResNet-50, Inception-V4, and traditional methods. Gaur et al. [20] combined AlexNet with Inception V3 and VGG16 to detect and classify six concrete defect types, achieving superior efficiency and accuracy compared to conventional techniques.

In 2014, Simonyan et al. [21] introduced VGGNet, featuring VGG-16 and VGG-19 architectures, which achieved second place in the ILSVRC image classification challenge. Its use of small 3×3 convolutional kernels improved feature extraction flexibility and reduced computational complexity. Gomez et al. [22] demonstrated the effectiveness of VGG in evaluating rolling bearing conditions through infrared thermal imaging. Que et al. [23] proposed a GAN-based data augmentation method and an improved VGG model for classifying asphalt pavement cracks, outperforming GoogLeNet, ResNet-18, and AlexNet. Chakrapani et al. [24] used transfer learning with AlexNet, VGG-16, GoogLeNet, and ResNet-50 for diagnosing clutch faults, with VGG-16 showing superior performance. Pandiyan et al. [25] developed a VGG-16-based EDCNN for detecting weld removal endpoints during sanding belt grinding, achieving effective predictions across different weld states.

In 2021, Ge et al. [26] introduced YOLO X, an improved architecture that replaced the traditional anchor-based framework with an anchor-free approach and separated classification and bounding box predictions with a Decoupled Head. Kong et al. [27] enhanced YOLO X with the s-mosica and kt-iou algorithms for printed solder paste defect detection, proving its adaptability to other industrial tasks. Feng et al. [28] developed a real-time YOLO X-based algorithm for detecting surface defects on oranges, improving detection efficiency with residual connections and cascaded networks. Wang et al. [29] proposed Yolo X-BTFPN for conveyor belt damage detection, using BTFPN and SimOTA to address imbalance and feature allocation issues, outperforming existing methods in reliability and convergence. Liu et al. [30] introduced a lightweight model based on attention mechanisms to improve wind turbine blade surface defect detection, enhancing YOLO X with bidirectional feature fusion for better recognition rates and faster detection.

In recent years, Transformer-based inspection methods have been widely used in industrial quality control due to their strong feature extraction, self-attention mechanisms, and ability to analyze complex patterns in visual and sequential data. They enhance defect detection, predictive maintenance, and anomaly detection. Pereira et al. [31] reviewed Transformer architectures for computer vision, discussing their strengths and limitations. Hütten et al. [32] compared CNN and Transformer-based models for railway freight car maintenance, showing comparable performance even with limited data. Wang et al. [33] explored Transformer applications in mechanical fault diagnosis, while Souza et al. [34] analyzed AI-driven Transformer thermal modeling. Ma et al. [35] reviewed Transformer-based anomaly detection methods, covering theoretical and practical aspects. Despite their accuracy, these methods face challenges such as high computational costs, data dependency, slow inference speed, generalization issues, and lack of standardization, limiting their real-time deployment in industrial settings [31,35].

### 2.3. Applications of Transfer Learning in Industrial Inspection

Transfer learning is essential in industrial inspection, enabling defect detection, quality control, and predictive maintenance with limited labeled data. By fine-tuning pre-trained deep learning models (e.g., ImageNet) for specific tasks, companies achieve higher accuracy with fewer data and lower training costs. Azari et al. [36] analyzed transfer learning in predictive maintenance within Industry 4.0. Bhuiyan and Uddin [37] reviewed vision-based defect detection using deep transfer learning with vibration acoustic sensor data. Yan et al. [38] highlighted deep transfer learning’s effectiveness in anomaly detection. Gaugel and Reichert [39] studied time series segmentation, showing how pre-trained models improve tasks like pump testing. Semitela et al. [40] explored transfer learning in vision-based surface defect inspection, comparing CNN architectures and illumination strategies. Despite its benefits, transfer learning faces challenges such as domain mismatch, high computational demands, lack of pre-trained industrial models, small dataset overfitting, explainability issues, and maintenance difficulties [36,40].

Transfer learning utilizes a model trained in one task to solve another, making it valuable for factories with frequent production line changes. Pan et al. [41] categorized transfer learning into classification, regression, and clustering while linking it to domain adaptation and multi-task learning. Liu et al. [42] combined transfer learning and data augmentation to train CNNs for injection molding defect detection, achieving high accuracy with minimal data. Xie et al. [43] integrated attention mechanisms with CNNs for solar cell defect detection, improving adaptation to new batches. Vinodhini et al. [44] applied CNNs and transfer learning for asphalt pothole detection, surpassing other deep learning methods. Kumar et al. [45] developed a semi-supervised transfer learning network for weld seam defect detection, addressing data imbalance and automating visual inspection. Gupta et al. [46] combined transfer learning with CNNs for casting defect analysis, reducing overfitting. Alqarni et al. [47] implemented a transfer-learning-based energy conservation model for autonomous vehicle navigation. This study applies a pre-trained model and limited target domain data to detect assembly defects in similar manual tool models.

Based on the above literature, it is evident that various industries are gradually shifting towards automated inspection, with recent research primarily employing deep learning techniques to develop detection systems and comparing different network models or methods under similar conditions. In the field of defect inspection, surface defects dominate the majority of research, while studies on assembly defects remain relatively scarce, often focusing on just one or two kinds of assembly anomalies. This study, however, explores multiple kinds of assembly anomalies. Furthermore, this research investigates defects in the assembly process of similar types of manual tools, which, in contrast to other studies focusing on single-product models, make this approach applicable to a wide range of products. By combining deep learning and transfer learning techniques, this study trains an assembly defect detection model, ultimately enabling accurate identification of the location and type of assembly defects.

## 3. Proposed Methods

This study aims to develop a system for inspecting assembly defects in manual tools by integrating computer vision and deep learning technologies. The overall research framework is divided into two main phases. The first phase involves training a pre-trained model, while the second phase uses the pre-trained model with transfer learning to adapt the system for manual tools of different specifications.

The first phase of this study consists of eight steps. Initially, experimental images are captured by placing the workpiece on a fixture and using a CCD camera. The captured images undergo preprocessing with Contrast-Limited Adaptive Histogram Equalization (CLAHE) to enhance contrast and make internal parts more distinguishable. Subsequently, image label files are created by manually annotating regions of interest. Features are then extracted using an AlexNet model to obtain image feature vectors, and the pre-trained model’s weight parameters are saved. Following this, Support Vector Machines (SVMs) are utilized to classify the foreground and background of various defect types. Defects are located by employing bounding box regression to identify their optimal positions. The phase concludes with the output of prediction results, creating a confusion matrix and calculating performance metrics to evaluate the model’s effectiveness.

The second phase builds upon the first and includes ten steps. The first three steps are identical to those in the first phase: capturing images, preprocessing them, and creating labeled datasets. After feature extraction, the weight parameters of the pre-trained model are imported. The network architecture is then modified, and the model is trained to fine-tune the updated weights. Steps six through nine mirror those from the first phase, involving the classification of defects, defect localization, and performance evaluation. The final step in this phase is a sensitivity analysis to explore the impacts of various ratios of training images in the target domain and different similarities of workpiece parts in training images on the model’s performance.

### 3.1. Image Capture

This study selected a swivel-head ratchet wrench as the sample for image capture. The workpiece was placed flat on a fixture during image capture since the shooting location was within a laboratory, as shown in Figure 5. A white paperboard was used as a background above the platform to reduce background complexity, and a direct-type ring light was employed as the auxiliary light source. A color CCD industrial camera was mounted at the center of the ring light to capture images. The light intensity was then adjusted to an appropriate level to ensure the internal components of the workpiece were clearly visible.

The workpiece itself can be divided into two main parts: the ratchet head and the handle. Due to the internal components, the surface of the ratchet head protrudes, causing it to be slightly tilted when placed on the platform. This tilt results in metallic reflections on the surface of the ratchet head during image capture. A fixture was designed and manufactured using 3D printing to address this issue, as shown on the right side of Figure 5. To prevent parts of the workpiece from being obscured during image capture, the use of fixtures is divided into two situations: capturing the front and back of the workpiece. Considering the possible locations of defects at each assembly station, the first and third assembly stations require capturing the front of the workpiece, while the second assembly station requires capturing both the front and back sides. When used, the fixture allows the workpiece to remain level during image capture, as illustrated in Figure 6.

### 3.2. Image Preprocessing

The experimental images captured in this study were processed using three different methods: histogram equalization (HE), adaptive histogram equalization (AHE), Contrast-Limited Adaptive Histogram Equalization (CLAHE) [48], and their Root Mean-Square Error (RMSE) and Peak Signal-to-Noise Ratio (PSNR) were calculated to compare the processed images. The evaluation metrics are defined as follows: 1. RMSE: This metric measures the difference between images and indicates image quality. A smaller RMSE indicates less difference and higher image quality, while a larger RMSE signifies greater differences and lower image quality. 2. PSNR: This metric assesses the distortion caused by noise or compression in an image, often comparing the original image with one affected by noise or compression. A higher PSNR indicates less impact from noise and better image quality, while a lower PSNR suggests the opposite. As shown in Table 2, among the three methods, the CLAHE method produced the smallest RMSE and the highest PSNR, making it the most effective in highlighting details without increasing background complexity. Therefore, this study selects CLAHE as the image preprocessing method.

### 3.3. Defect Detection System in Manual Tool Assembly Abnormalities

After completing image preprocessing, the next step involves creating image label files by manually annotating regions of interest (ROI). These annotations are then used for detection with a neural network model. In subsequent chapters, definitions of ROI, bounding boxes, and the generation of candidate regions will be explained. This study employs an R-CNN model combined with a transfer learning training process. R-CNN is a deep learning model designed for object detection, with the primary goal of identifying specific objects in images and accurately locating their bounding boxes. R-CNN uses “selective search” to generate multiple candidate regions that might contain objects. Each candidate region is then resized to a fixed size and fed into a convolutional neural network for feature extraction. The extracted features are subsequently classified using an SVM, and bounding box regression is applied to refine the localization for improved detection accuracy. R-CNN was a groundbreaking approach that introduced deep learning to object detection, but its efficiency is limited by the large number of candidate regions that need to be processed [12].

R-CNN and AlexNet are often combined to improve object detection tasks, utilizing AlexNet as the feature extractor within the R-CNN pipeline. The R-CNN with AlexNet detects objects by first generating region proposals using selective search, which identifies potential bounding boxes. Each region is resized to 227 × 227 pixels and passed through AlexNet, a pre-trained CNN, to extract deep features. These features are then classified using SVMs to determine object categories. To improve localization, bounding box regression adjusts the position and size of predicted boxes using learned offsets. Finally, it eliminates redundant detections, refining the final predictions. This approach utilizes AlexNet’s deep feature representation to enhance detection accuracy, making it widely used in industrial inspection and automated vision systems. Figure 7 shows the detailed workflow of R-CNN with the AlexNet model.

Figure 8 illustrates the training procedure for network architecture using R-CNN with the AlexNet model. The process begins with image labeling, where each object in the dataset is annotated with bounding boxes and class labels. Selective search generates approximately 2000 region proposals per image, which are then resized to 227 × 227 pixels and fed into AlexNet, a pre-trained CNN, for feature extraction. These extracted features are classified using SVMs to identify object categories. To enhance localization accuracy, a bounding box regression model learns offsets to refine the predicted bounding boxes. The model outputs include class labels, confidence scores, and adjusted bounding box coordinates. Additionally, AlexNet can be fine-tuned on the target dataset for improved performance. The final model comprises AlexNet’s CNN parameters, SVM weights, and bounding box regression coefficients, ensuring high-precision object detection for industrial inspection and automated vision systems.

#### 3.3.1. Explanation of Region Definitions

The ROI refers to the manually marked area, which represents the region of the object to be detected. The region proposal (RP) is generated by the system using the selective search method, identifying potential areas that may contain defects. The bounding box (BB) represents the final area identified by the system as containing a defect. The definitions of ROI, RP, and BB are shown in Figure 9.

#### 3.3.2. Feature Extraction of ROI Images

AlexNet is a deep convolutional neural network model designed for image classification tasks. It was the first to demonstrate the advantages of deep learning on large-scale image datasets through its success in the ImageNet classification challenge [16]. AlexNet’s architecture includes multiple convolutional layers, pooling layers, and fully connected layers. It employs ReLU activation functions and Dropout techniques to accelerate convergence and prevent overfitting. The model also utilizes data augmentation and GPU acceleration during training, significantly improving classification accuracy. The success of AlexNet paved the way for the widespread adoption of deep learning in various computer vision tasks [20].

The key connection between R-CNN and AlexNet lies in the feature extraction process. R-CNN utilizes AlexNet as a feature extractor, feeding candidate regions into AlexNet to extract high-level deep features, which serve as the foundation for subsequent classification and bounding box regression. By utilizing AlexNet, which is pre-trained in ImageNet, R-CNN employs transfer learning, significantly reducing the data and computational resources required for training. This combination pioneered the application of deep learning models in object detection and laid the groundwork for subsequent improved models, such as Fast R-CNN and Faster R-CNN.

After preprocessing the image, the Image Labeler tool in MATLAB R2021a was used to label the defect areas in the image, and the labeling method used was drawing bounding boxes. The selective search algorithm is used to generate multiple candidate regions for the input image, and the AlexNet model is used to extract features from each candidate region. Since the input image size limitation for the AlexNet model is 227 × 227, and the candidate regions generated by the selective search algorithm have varying sizes, the candidate regions must first be resized to 227 × 227 before they can be processed through the convolutional layers to extract feature vectors.

#### 3.3.3. Defect Classification of SVM

After extracting feature vectors using AlexNet, each candidate region’s feature vector is fed into separate SVM classifiers for each class. These classifiers score each candidate region, representing the probability that the region belongs to that specific class. Based on the scores from each SVM classifier, the class with the highest score is selected as the predicted class for the candidate region. Typically, in addition to the individual class SVMs, there is also an additional background SVM classifier. If the candidate region does not belong to any class, it is classified as background.

#### 3.3.4. Adjusting the Location of the Predicted Candidate Region

Bounding box regression is a technique used to correct and fine-tune the position of bounding boxes. It calculates the positional offset between the ground truth bounding box and the predicted bounding box using a linear regression model, allowing each candidate region to generate corresponding labels. The generated labels and feature vectors are used to train the regression model, enabling it to learn how to correct the position of the predicted bounding boxes. During testing, the trained regression model is used to predict the positional correction, and this correction is applied to the predicted bounding boxes to obtain more accurate defect locations.

Bounding box regression is a key step in defect detection tasks, refining initially predicted bounding boxes to better match the true defect locations [11]. Bounding box regression refines predicted bounding boxes *B* = (*x*, *y*, *w*, *h*) to better match ground-truth defects *B_gt_ =* (*x_gt_*, *y_gt_*, *w_gt_*, *h_gt_*) by learning transformations, where (*x*, *y*) are the center coordinates of the bounding box and (*w*, *h*) are the width and height of the bounding box. Instead of predicting *B*′ directly, the R-CNN model learns four offset values:(1)$$t_{x}=\frac{x_{gt}-x}{w},\;{\;t}_{y}=\frac{y_{gt}-y}{h}$$(2)$$t_{w}=log\frac{w_{gt}}{w},\;{\;t}_{h}=log\frac{h_{gt}}{h}$$These offsets adjust position and scale in a normalized, scale-invariant manner.

R-CNN models bounding box regression as a linear transformation. A fully connected layer predicts the offsets using extracted feature vector *ϕ*(*B*):(3)$${\hat{t}}_{x}=W_{x}\cdot \phi \left(B\right)+b_{x},\;{\hat{t}}_{y}=W_{y}\cdot \phi \left(B\right)+b_{y}$$(4)$${\hat{t}}_{w}=W_{w}\cdot \phi \left(B\right)+b_{w},\;\;{\hat{t}}_{h}=W_{h}\cdot \phi \left(B\right)+b_{h}$$
where *W* is the weight matrix learned during training, *b* is the bias term. Thus, the feature extraction network, AlexNet, generates a feature vector for the defect proposal, and a fully connected layer predicts the bounding-box correction terms.

Bounding box regression is trained using Smooth L1 Loss:(5)$$L_{reg}\left(t,\;\hat{t}\right)=\sum_{i\in \left\{x,\;y,\;w,\;h\right\}}Smooth\;L1(t_{i}-{\hat{t}}_{i}),$$(6)$$Smooth\;L1\left(x\right)=\left\{\begin{array}{c} 0.5x^{2},\;\;if\;\left|x\right|<1\\ \left|x\right|-0.5,\;\;otherwise \end{array}\right.$$This balances small and large errors for robust learning and improves stability in learning bounding box refinements. Once the model is trained, the refined bounding box *B*′ is computed as(7)$$x^{\prime }=x+{\hat{t}}_{x}\;w,\;y^{\prime }=y+{\hat{t}}_{y}\;h$$(8)$$w^{\prime }=w\cdot e^{{\hat{t}}_{w}},\;\;h^{\prime }=h\cdot e^{{\hat{t}}_{h}}$$This step adjusts the bounding box position and size so that it aligns more closely with the actual defect region. The bounding box regression approach follows these steps: (1) refine initial defect proposals into more accurate bounding boxes, (2) use linear regression to learn position and size offsets, (3) apply Smooth L1 Loss for stable training, and (4) adjust bounding boxes at inference using learned transformations. This method is widely used in industrial defect detection, enhancing the accuracy of automated inspections.

#### 3.3.5. Procedures and Parameter Adjustment for the Transfer Learning Model

Transfer learning accelerates the development of defect detection models for new products by utilizing pre-trained models from existing products. Since products within the same industry share common features, transfer learning optimizes and speeds up model adaptation without requiring full retraining. This is especially beneficial in assembly lines with diverse products and frequent changes. It enhances feature extraction efficiency, reduces training time, and mitigates overfitting due to limited data. However, it may compromise accuracy and generalization. The effectiveness of transfer learning depends on the similarity between pre-trained weights and the target dataset, with closer feature alignment yielding better results [3,41].

Transfer learning involves applying the knowledge learned from the source domain task to the target task, which is a related but different task. The training and testing framework of the transfer learning scheme applied in this study is shown in Figure 10. The training and testing stages are connected through transfer learning, where a pre-trained model from a source domain provides initial weight parameters to an R-CNN model for defect detection. During training, the model is fine-tuned using target domain data, adjusting feature extraction layers to learn domain-specific characteristics. The trained model extracts features, classifies defects using SVMs, and refines bounding boxes through regression. In testing, new images undergo the same preprocessing and selective search, and the trained model generates defect classifications and refined bounding boxes. This transfer learning process enables effective defect detection by utilizing prior knowledge while adapting to new inspection tasks.

The transfer learning method in this study follows these steps:Build the target R-CNN model—develop an R-CNN model for the target domain with newly initialized weights. The architecture is based on AlexNet, consisting of five convolutional layers and three fully connected layers.Transfer weights from the source model—use the final trained weights from a pre-trained model in the source domain as initial weights for the target domain model.Freeze shallow convolutional layers—keep the weights of the first two convolutional layers (for example) fixed during training. These layers capture low-level features such as edges and lines, which are generally transferable across domains.Fine-tune deep convolutional layers—allow the last three convolutional layers (for example) to update during training. These layers extract task-specific features, such as levers and springs, improving adaptation to the new dataset.Modify fully connected layers—adjust the parameters of the fully connected layers to ensure their output matches the number of defect types identified in this study.Train the target domain model—train the model using the target domain dataset, fine-tuning weight parameters with limited data. Transfer learning enhances defect detection by utilizing pre-trained features, improving accuracy in similar product inspections.

Figure 11 and Figure 12 illustrate the AlexNet model architecture with transfer learning and the step-by-step process of transfer learning applied in this study.

## 4. Experiments and Results

This study utilizes various equipment to capture images of assembly defects, including a personal computer (specifications: CPU: AMD Ryzen™ 9 5900HS 4.6 GHz, 32 GB RAM, GPU: NVIDIA GeForce^®^ RTX 3060, operating system: Windows 10), a high-resolution CCD camera, a specialized Basler lens, and fixtures for positioning workpieces. During the imaging process, a ring light is used to enhance the capture of assembly defect images.

This study utilizes MATLAB R2021a application software to develop a system for inspecting the assembly of manual tools with various specifications. The user interface is shown in Figure 13. The left side features settings for parameter adjustments, such as image preprocessing, product selection, and model selection. The right side displays the execution results, including the original image, enhanced image, inspection image, detection type, and detection location. Finally, the system provides a statistical count of defects categorized as missing parts, misplacements, and extra object anomalies.

### 4.1. Performance Evaluation Metrics for Assembly Defect Detection

This study evaluates the performance of assembly defect detection by analyzing classification results categorized as good or defective products. Key metrics include recall, precision, and correct classification rate (CR, accuracy). The precision and recall are further used to calculate the F1-Score for overall system performance. If the dataset is balanced, CR becomes a more reliable evaluation metric for multi-class defect classification. Since no class dominates the dataset, CR fairly represents the model’s overall performance without being biased toward any particular class. In the experiments of this study, since the datasets are all balanced, CR is used as the primary metric to be measured. For defect identification, correct classification requires the defect type to be accurate and the bounding box overlap to exceed 50%. Misclassifications occur if the defect type is correct but the location is wrong or if the defect location is correct but the type is incorrect.

### 4.2. Setting Different Transfer Learning Ways for Detection Methods

Two methods are used to implement the transfer learning method. One is to freeze the lower part of the hidden layer, using the front-stage feature extraction module to do fine-tuning training with the data backstage. For example, freeze most of the convolutional layers close to the input of the pre-trained model and train some of the convolutional and fully connected layers close to the output layer. The other is to freeze the high-level part of the hidden layer and retrain the feature extraction module that matches the data from the lower level. Finally, the output neurons of the last layer of the model will be modified to match the number of categories of the new data. During the fine-tuning process, a smaller learning rate must be used to train the network. If the amount of data in the data set is insufficient, we can only train the last layer; if the number of data sets is medium, we can freeze the weight of the first few layers of the pre-trained network.

The detection method is configured using transfer learning, with AlexNet selected as the adjusted network model. Experiments are conducted to observe which configuration yields better detection performance by freezing convolutional layers responsible for extracting either low-level or high-level features within the five convolutional layers. Ten different transfer learning approaches are evaluated to determine their classification accuracy, selecting the most effective configuration. Using a reduced sample set from the first station, the model is trained with 450 source domain images and 450 target domain images. Table 3 presents the differences among the ten transfer learning approaches, while Table 4 provides details on the parameter settings of three operations, freezing (in blue), fine-tuning (in green), and modification (in red), for transfer learning. Figure 11, Figure 14 and Figure 15 indicate the network architecture diagram of the TL4, TL5, and TL10 transfer learning methods, respectively.

Table 5 presents the detection results for ten transfer learning approaches. As shown in the line chart of Figure 16, in terms of correct classification rates, the TL4 approach—freezing the first two convolutional layers, fine-tuning the subsequent three convolutional layers, and modifying the output layer—achieves the best detection results. Subsequently, an efficiency comparison was conducted for the top three performing transfer learning approaches: TL4, TL5, and TL10. As illustrated in the bar chart of Figure 16, TL4 not only required the least amount of time but also delivered the highest performance. Therefore, this study adopts TL4 as the chosen transfer learning approach.

### 4.3. Parameter Settings for Network Models of Detection Methods

To enhance the accuracy and generalization capability of the proposed detection method, this study adjusts various parameters. These adjustments include settings for the optimizer, learning rate, batch size, and training epochs. By evaluating the correct classification rate (CR), the study identifies better parameter configurations. Experiments are conducted using small samples from the first assembly station, with all images sized at 1280 × 1024 pixels. The target domain dataset consists of 150 training images and 75 testing images.

The network model parameters are adjusted, with AlexNet selected as the model for parameter tuning. The default parameter values are as follows: optimizer: SGDM; learning rate: 0.001; training batch size: 32; and training epochs: 10. Subsequently, experiments are conducted using different combinations of parameters in the order of optimizer, learning rate, training batch size, and training epochs. The experimental results are compared to identify the better parameter settings.

The better parameter settings for the detection model in this study are shown in Table 6. Experiments were conducted using different combinations of parameters in the order of optimizer, learning rate, training batch size, and training epochs. Small samples were used for testing, and the results were compared to determine the best configuration. These parameter settings will be applied in subsequent experiments with larger sample sizes.

### 4.4. Performance Evaluation of Detection Methods for Assembly Anomaly Kinds and Defect Types

When constructing image recognition models, a common issue encountered is poor recognition accuracy due to insufficient training samples or an imbalance in the number of samples between different categories. When the training images are too few, the model struggles to identify features during the training phase, resulting in insufficient features for classification or recognition during testing, ultimately leading to incorrect recognition. This study proposes an R-CNN model that combines deep learning and transfer learning methods. By training the model with a small dataset of highly similar images, it achieves good results even with a limited dataset. The large sample size and reduced sample size of the new product in the two datasets are shown in Table 7. The domain ratio is the ratio of the number of images in the source domain to that of the target domain during the training and testing stages for each defect type.

#### 4.4.1. Performance of AlexNet Model with and Without Transfer Learning

Typically, building a new model from scratch requires many samples to support its training. The objectives of this study’s experiments are (1) to investigate whether training a model using transfer learning yields superior results and (2) to examine the effectiveness of using transfer learning to train a model with a limited number of samples.

The effectiveness of all models is primarily evaluated using the CR metric as the main criterion due to the balanced dataset. Based on the data presented in Figure 17, the AlexNet model combined with transfer learning demonstrates a higher CR than the R-CNN model initialized with entirely new weights when applied to the dataset of a new product. Additionally, its efficiency is superior. A comparison of the detection efficiency for each workstation across the three stations is illustrated in Figure 18.

Next, the study examines the effectiveness of training models using transfer learning with a reduced sample size of new products. Since the large-sample experiment utilized 50 images per class, the reduced-sample experiment was conducted using 30 images per class—approximately a 40% reduction in new products. The experimental results conducted in three assembly stations show the AlexNet model with transfer learning consistently achieved higher correct classification rates compared to the R-CNN model initialized with entirely new weights, even with a 40% reduction in sample sizes of new products. The changes in effectiveness and efficiency for the three workstations are illustrated in Figure 19 when reducing 40% in sample sizes of new products by using the AlexNet model with and without transfer learning. Even with a 40% reduction in sample sizes for new products, the AlexNet model maintained a classification accuracy of 98.66% at the first station. Compared to R-CNN, it showed a 9% improvement in average effectiveness (changes in CR values) and a 26% increase in efficiency (changes in training time) across all stations.

#### 4.4.2. Comparison with Other Models of Transfer Learning

In the CNN family, VGGNet is a model developed by increasing the depth of CNNs. Due to its structural similarity to AlexNet and its strong performance in transfer learning tasks, it is often used as a benchmark for comparison. Based on the reduced-sample experiments from the previous section, this study conducts small-sample experiments for each workstation to determine whether the AlexNet model or the VGG16 model is better suited for this task when paired with transfer learning.

YOLO is currently one of the most popular methods in object detection algorithms. Unlike R-CNN, YOLO is a single-stage object detection algorithm. Compared to the two-stage object detection approach of R-CNN, YOLO reduces the computational complexity of the model, offering significant advantages in speed. However, this comes at the cost of some accuracy differences. We compare the proposed model’s effectiveness and efficiency with the YOLO X model to evaluate whether YOLO X is better suited for the detection tasks in this study.

Table 8 summarizes the effectiveness of different detection methods across the three workstations. As shown in Figure 20, when considering effectiveness alone, the evaluation metrics (recall, precision, F1-score, and correct classification rate CR) of VGG16 combined with transfer learning is slightly higher than those of other detection methods, followed closely by AlexNet with transfer learning, with only about a 1% difference from VGG16. Table 9 lists the training time and testing time for different network models at each assembly station. Figure 21 reveals the comparison chart of training and testing time for different network models at each station. As shown in Figure 21, AlexNet with transfer learning demonstrates the highest efficiency among the four methods in the training stage, while VGG16 exhibits the lowest efficiency in both the training and testing stages. Based on these results, AlexNet with transfer learning emerges as the optimal choice among the four detection methods when both effectiveness and efficiency are considered.

### 4.5. Robustness Analysis of the Proposed Method

#### 4.5.1. The Impact of the Ratio of Training Images in the Target Domain to the Source Domain on Detection Effectiveness

This study investigates the impact of the proportion of training images from the source domain and target domain on the detection effectiveness when using the AlexNet model combined with transfer learning. The objective is to identify the optimal ratio between the source and target domains that yields the best detection performance. The total number of experimental images is 1600, with the number of images for defect-free and defective products at each station shown in Table 10. The ratio of training to testing images is 2:1.

Figure 22 presents the detection results when training the AlexNet model with transfer learning using target domain training image proportions of 83%, 67%, 50%, 33%, and 17%, respectively. The curve in Figure 22 shows that as the proportion of target domain images increases, the detection effectiveness improves. Conversely, a lower proportion of target domain images results in fewer training images, reducing the training execution time. When the target domain image proportion reaches 50% or more, the CR value exhibits minimal or no significant change. Thus, it is concluded that a target domain training proportion of 50% is optimal for this detection model.

#### 4.5.2. The Impact of Different Similarities of Parts in Training Images on Detection Effectiveness

This study examines whether the similarity of assembled components affects detection performance. The number of components and shared components across the three workstations are counted. By calculating the component similarity for each station and using the experimental data from the previous section to compute the average correct classification rates, the results are summarized in Table 11. As shown in Figure 23, higher component similarity correlates with better detection performance. Since the similarity between the first and second stations does not vary significantly, their detection performance also shows minimal differences. Despite the third station having only 17% component similarity, it achieves a correct classification rate of 89%. The results indicated that the proposed method reduces training samples by 50% at 50% similarity, requiring only half the original samples while improving effectiveness by 13.06% and efficiency by 5.31%.

The performance of transfer learning is highly dependent on the similarity between the source and target domains. When similarity is very low (<20%), the model struggles to transfer useful features, leading to poor accuracy and requiring extensive fine-tuning, which reduces efficiency. As similarity increases to 50%, shared features improve accuracy and efficiency, though a significant amount of target domain data is still needed. With moderate similarity, the pre-trained features align well with the new task, boosting accuracy and efficiency while reducing the need for additional data. At high similarity (≥80%), transfer learning is highly effective, achieving near-optimal accuracy and efficiency with minimal fine-tuning. This trend highlights that greater similarity between domains enhances knowledge transfer, reducing training time, data requirements, and computational costs while improving detection performance.

The performance of transfer learning significantly improves as the similarity between the source and target domains increases due to enhanced feature reusability, reduced overfitting, and more efficient fine-tuning. When the source and target domains share structural characteristics, textures, or patterns, the pre-trained model can effectively utilize existing feature representations, minimizing the need for extensive retraining. This reduces the risk of overfitting, particularly when labeled data in the target domain is limited, ensuring better generalization. Additionally, as domain similarity increases, fewer network layers require fine-tuning, leading to faster convergence, lower computational costs, and improved stability in training. Therefore, maximizing domain similarity enhances transfer learning efficiency, allowing models to retain valuable pre-trained knowledge while adapting quickly and effectively to new tasks.

AlexNet can maintain high accuracy even with reduced sample sizes due to its hierarchical feature extraction, pre-training on large datasets, and effective regularization techniques. As a model pre-trained on ImageNet, AlexNet learns robust low- and mid-level features, such as edges, textures, and shapes, which can be transferred to new tasks with minimal labeled data. This significantly reduces training time and computational cost since only the later layers require fine-tuning. Additionally, using a pre-trained network mitigates overfitting, improving generalization. AlexNet’s convolutional architecture enhances feature extraction, making it effective for various tasks despite newer models like ResNet offering improved accuracy. For applications with limited data and computational resources, AlexNet remains a viable and efficient choice.

## 5. Concluding Remarks

This study presents an inspection system for manual tool assembly, specifically focusing on defect detection and classification in flex-head ratchet wrenches as an alternative to traditional inspection methods. Utilizing a deep learning model with transfer learning, the system identifies and classifies assembly defects across similar tools. The effectiveness of the AlexNet model with transfer learning is compared against other models at three assembly stations, demonstrating its superiority as the optimal inspection solution. Experimental results show that at the first station, which had the highest variety of defects, the AlexNet model achieved a classification accuracy of 98.67%, outperforming the R-CNN model with randomly initialized weights at all stations. Even with a 40% reduction in sample size for new products, AlexNet maintains a classification accuracy of 98.66% at the first station. Compared to R-CNN, it achieves a 9% improvement in average effectiveness and a 26% increase in efficiency across all stations.

This study’s sensitivity analysis examines factors influencing detection effectiveness, including the ratio of source and target domain training images and part similarity. The results show that the proposed method reduces training samples by 50% when part similarity is at 50%, requiring only half the original samples while improving effectiveness by 13.06% and efficiency by 5.31%. However, several challenges remain for future research, including integrating the proposed vision system with robotics, managing multiple defect types on a single workpiece, inspecting multiple workpieces simultaneously, and analyzing part similarity between source and target domains to estimate the required reduction in target domain images.

## Figures and Tables

**Figure 1 sensors-25-01645-f001:**
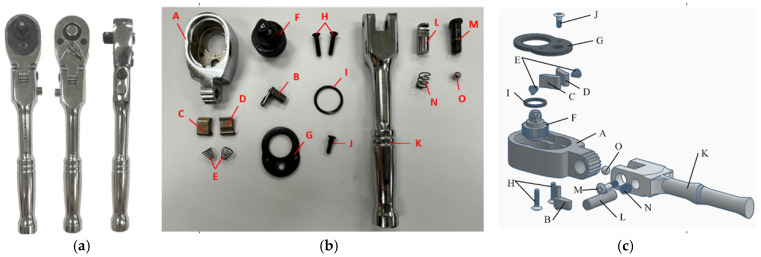
Assembly structures of 1/4″ DR 72T flex-head ratchet wrench: (**a**) three views (front, back, side) of an assembled product; (**b**) physical parts diagram; (**c**) exploded view drawing.

**Figure 2 sensors-25-01645-f002:**
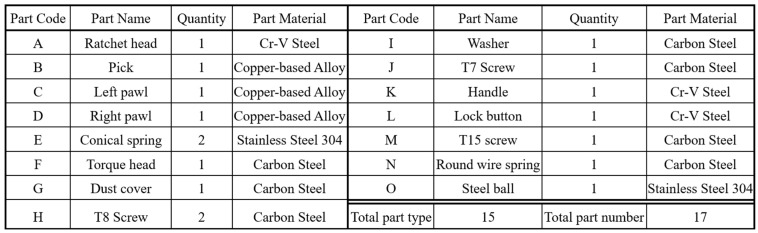
A list of assembly parts of 1/4″ DR 72T flex-head ratchet wrench.

**Figure 3 sensors-25-01645-f003:**
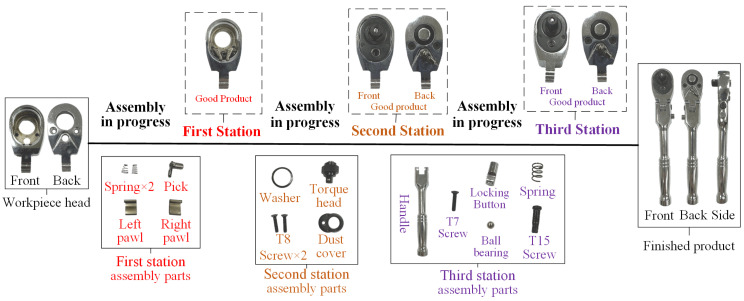
A diagram illustrating the parts required for each station in the flex-head ratchet wrench assembly process.

**Figure 4 sensors-25-01645-f004:**
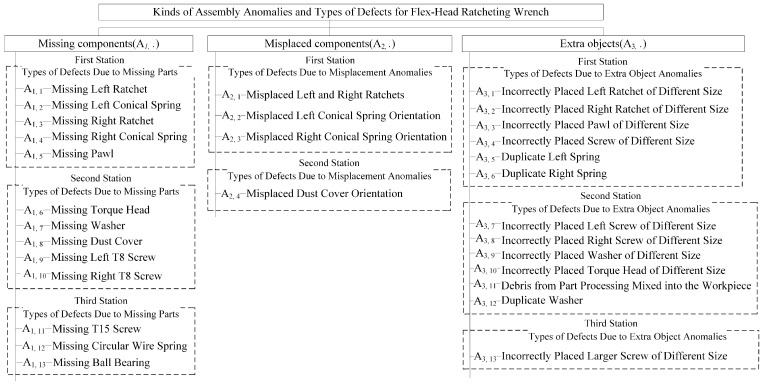
A summary chart depicting the correlation between assembly anomalies and defect types in flex-head ratchet wrenches.

**Figure 5 sensors-25-01645-f005:**
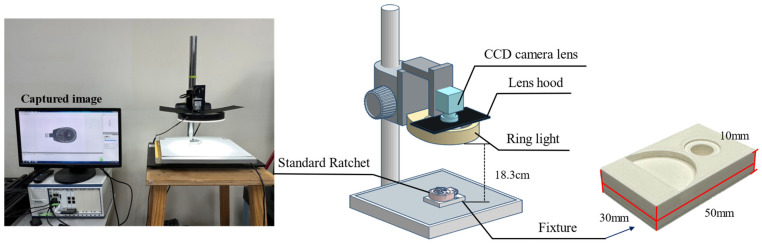
Hardware configuration for image capture: a photograph alongside its schematic diagram.

**Figure 6 sensors-25-01645-f006:**
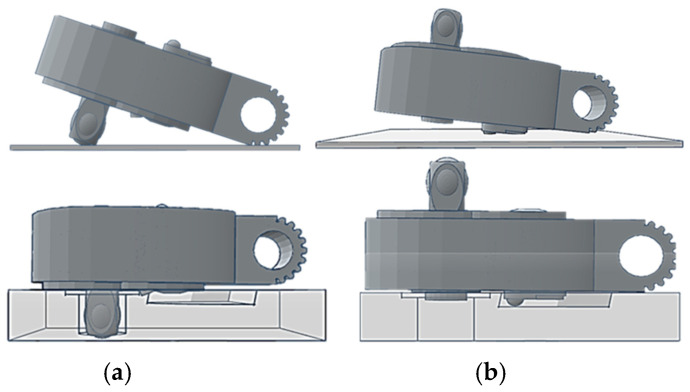
Illustration of the comparison of the test workpiece with and without using a fixture: (**a**) when capturing the front side of a workpiece, (**b**) when capturing the back side of a workpiece.

**Figure 7 sensors-25-01645-f007:**
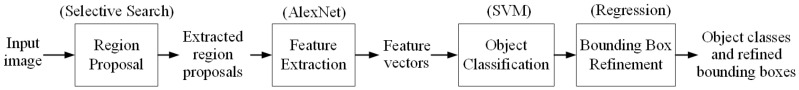
The detailed workflow of R-CNN with the AlexNet model.

**Figure 8 sensors-25-01645-f008:**
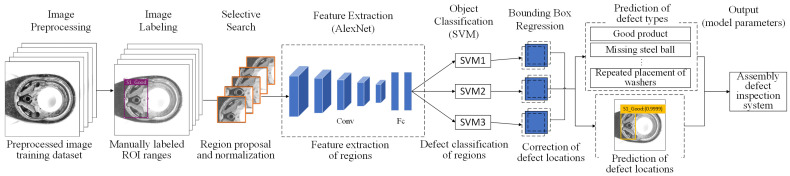
Training procedure for network architecture using R-CNN with AlexNet model.

**Figure 9 sensors-25-01645-f009:**
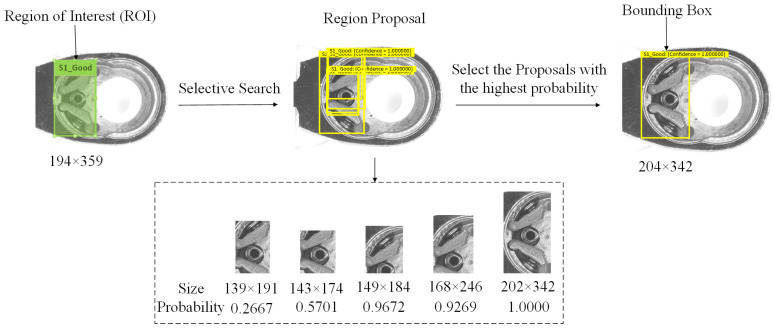
Definitions of region of interest, region proposal, and bounding box.

**Figure 10 sensors-25-01645-f010:**
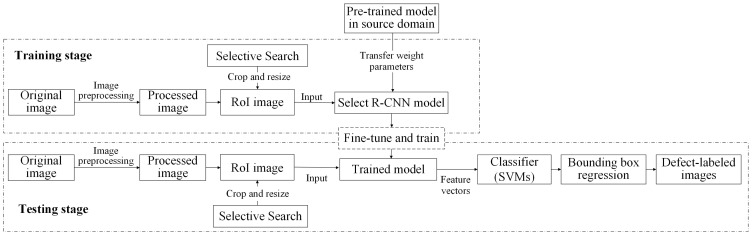
Training and testing framework of the transfer learning scheme in this study.

**Figure 11 sensors-25-01645-f011:**
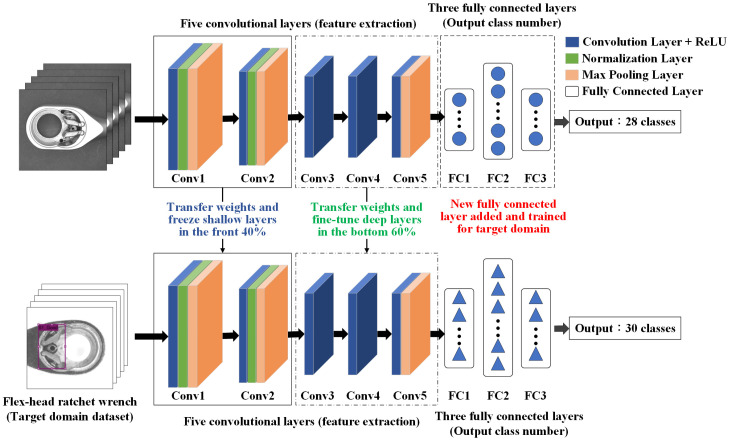
An example of an architecture diagram of the AlexNet model with transfer learning.

**Figure 12 sensors-25-01645-f012:**
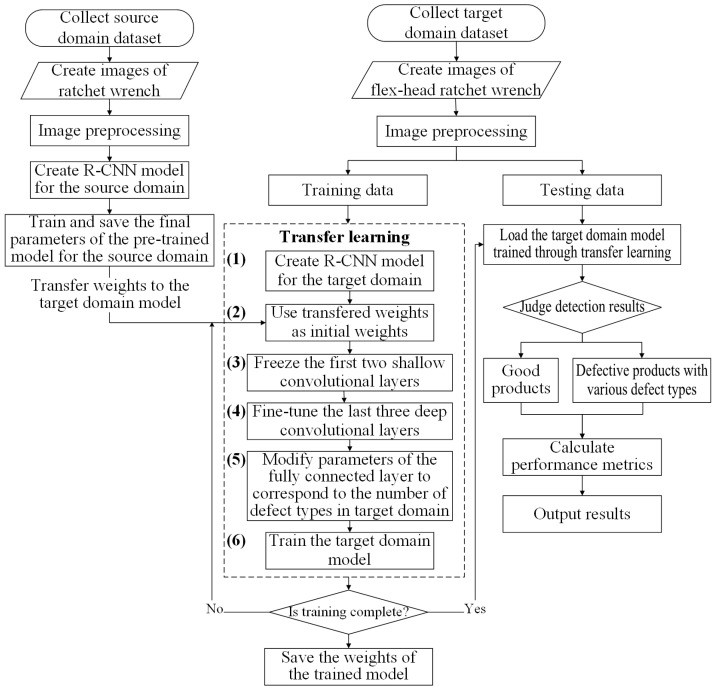
An example of the step diagram of transfer learning applied in this study.

**Figure 13 sensors-25-01645-f013:**
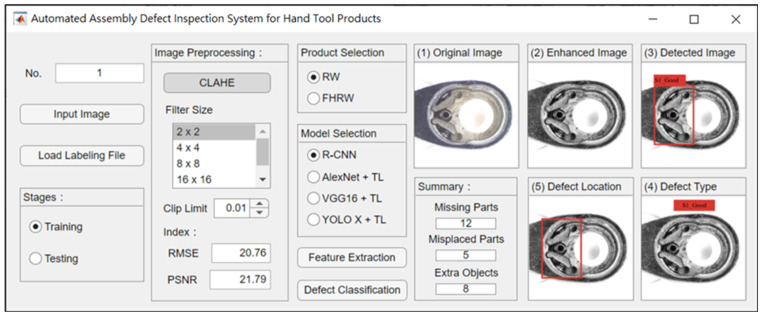
User interface of the detection system developed in this study.

**Figure 14 sensors-25-01645-f014:**
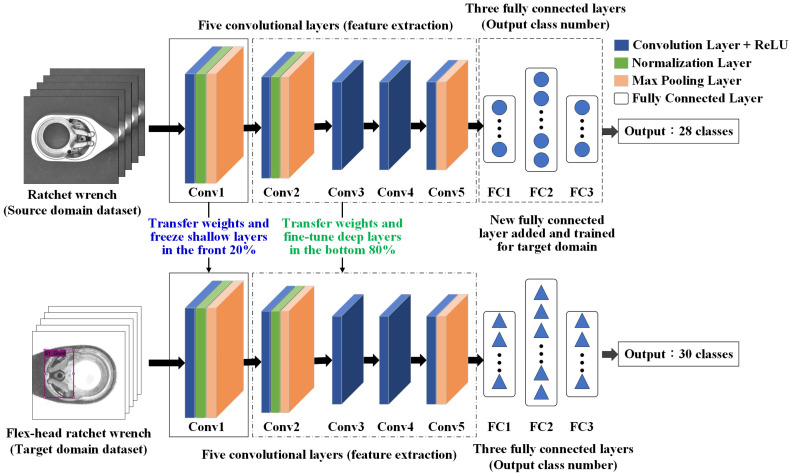
Network architecture diagram of TL5 transfer learning method.

**Figure 15 sensors-25-01645-f015:**
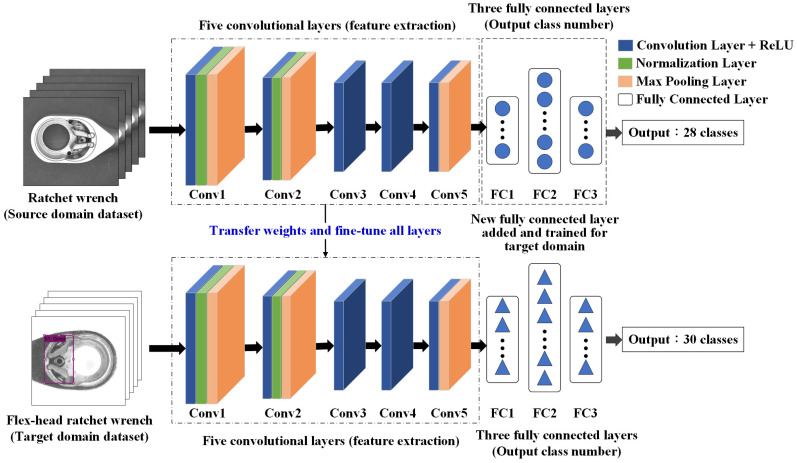
Network architecture diagram of TL10 transfer learning method.

**Figure 16 sensors-25-01645-f016:**
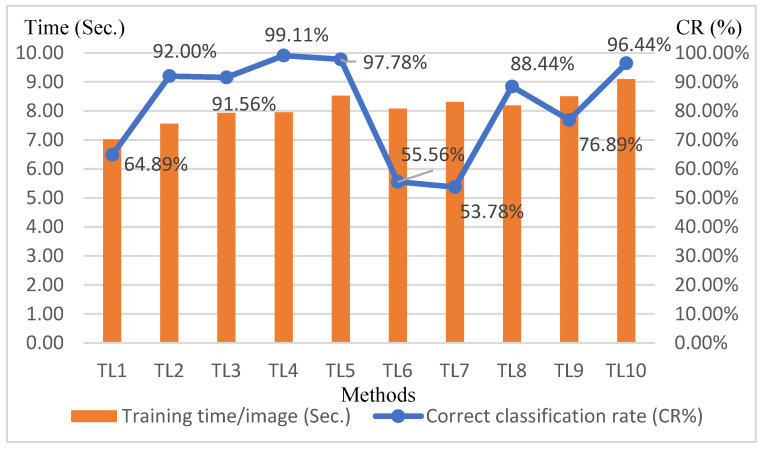
Combination chart of correct classification rates and training time per image for different transfer learning methods.

**Figure 17 sensors-25-01645-f017:**
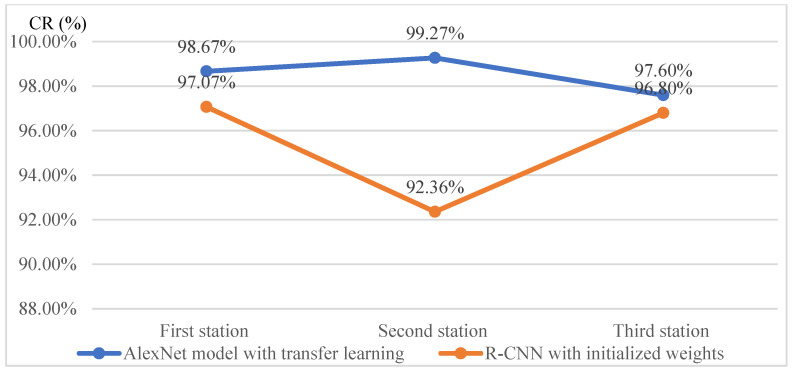
Comparison chart of correct classification rates for large new samples using the AlexNet model with and without transfer learning at each station.

**Figure 18 sensors-25-01645-f018:**
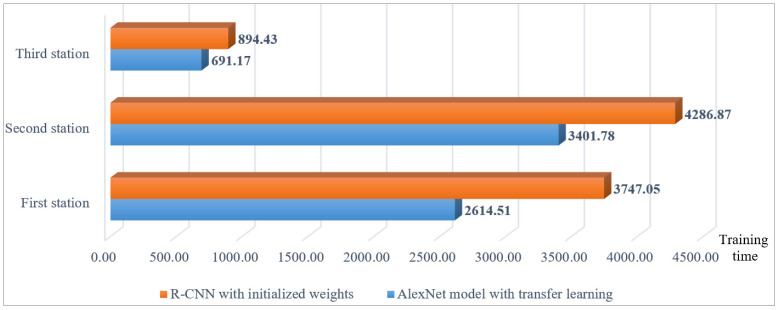
Comparison chart of training time for large new samples using the AlexNet model with and without transfer learning at each station.

**Figure 19 sensors-25-01645-f019:**
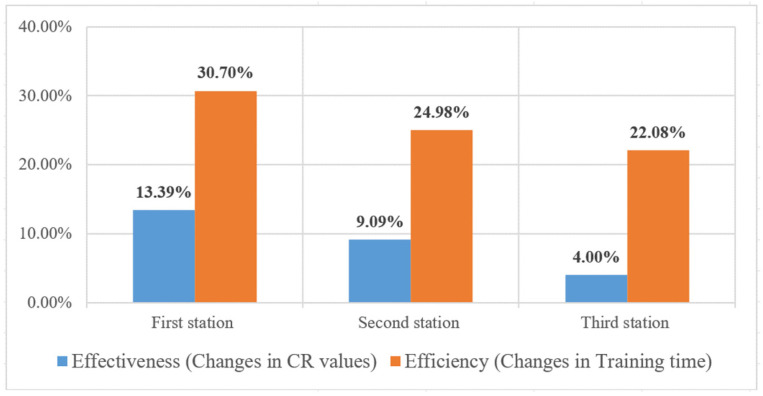
Changes in training time and testing CR values of each station when reducing 40% in sample sizes of new products by using the AlexNet model with and without transfer learning.

**Figure 20 sensors-25-01645-f020:**
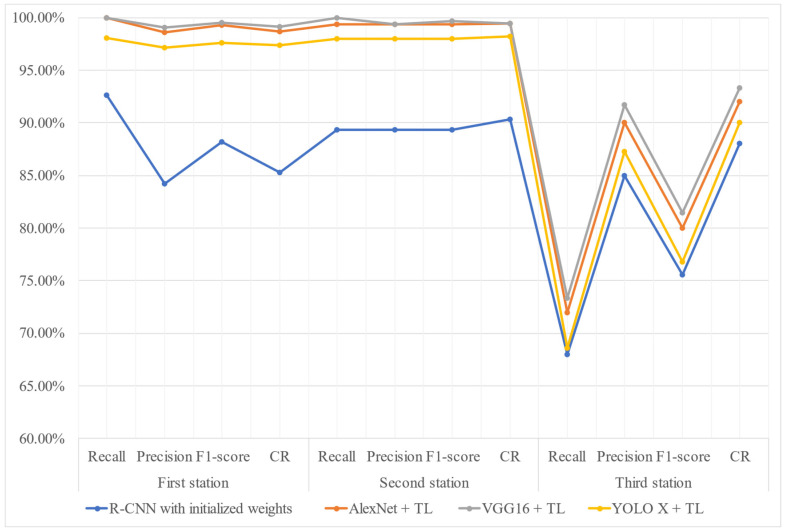
Comparison chart of effectiveness metrics for different network models at each assembly station.

**Figure 21 sensors-25-01645-f021:**
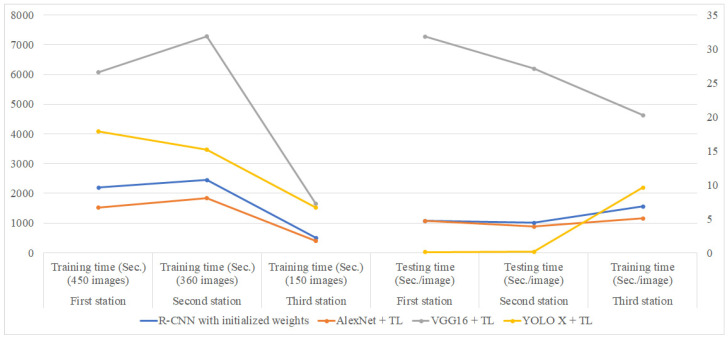
Comparison chart of training and testing time for different network models at each station.

**Figure 22 sensors-25-01645-f022:**
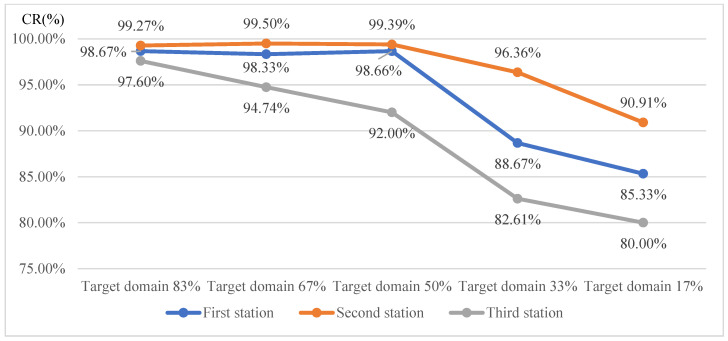
Line chart of correct classification rates of each station under different proportions of training images in the target domain using AlexNet with transfer learning.

**Figure 23 sensors-25-01645-f023:**
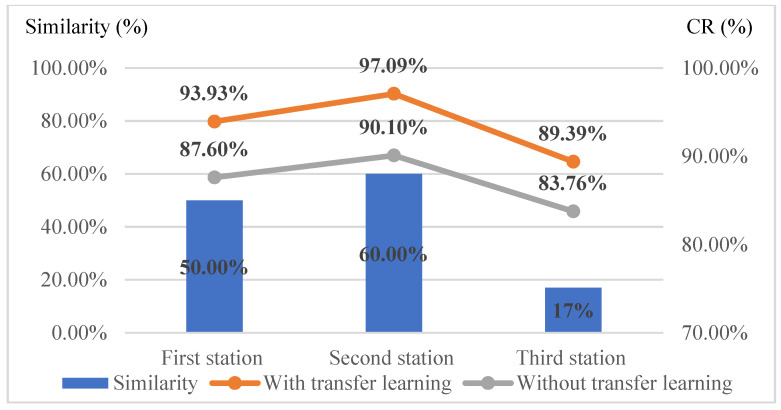
Line plots of part similarity and averages of correct classification rates at different stations using AlexNet with and without transfer learning.

**Table 1 sensors-25-01645-t001:** Comparison table of parts and defect-related items of two similar ratchet wrenches.

Domains	Source Domain (Previous Study [4])	Target Domain (This Study)
Product	Ratchet wrench	Flex-head ratchet wrench
Part number	12	17
Anomaly kinds	4	3
Defect types	28	30
Number of common parts (Similarity %)	7 (58%)	7 (41%)

**Table 2 sensors-25-01645-t002:** Comparison table of images before and after using three different preprocessing methods.

Preprocessing Method	None	HE Method	AHE Method	CLAHE Method
Effect after processing	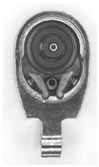	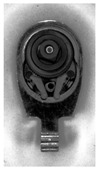	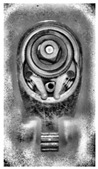	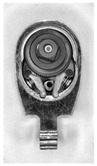
RMSE	-	58.8113	81.4831	21.2543
PSNR	-	24.4591	25.7188	27.6681

**Table 3 sensors-25-01645-t003:** Differences in the network architecture of ten transfer learning methods.

Transfer Learning Methods	TL1 (100%)	TL2 (+80%)	TL3 (+60%)	TL4 (+40%)	TL5 (+20%)	TL6 (−80%)	TL7 (−60%)	TL8 (−40%)	TL9 (−20%)	TL10 (0%)
Conv1	Freezing	Freezing	Freezing	Freezing	Freezing	Fine-tuning	Fine-tuning	Fine-tuning	Fine-tuning	Fine-tuning
Conv2	Freezing	Freezing	Freezing	Freezing	Fine-tuning	Freezing	Fine-tuning	Fine-tuning	Fine-tuning	Fine-tuning
Conv3	Freezing	Freezing	Freezing	Fine-tuning	Fine-tuning	Freezing	Freezing	Fine-tuning	Fine-tuning	Fine-tuning
Conv4	Freezing	Freezing	Fine-tuning	Fine-tuning	Fine-tuning	Freezing	Freezing	Freezing	Fine-tuning	Fine-tuning
Conv5	Freezing	Fine-tuning	Fine-tuning	Fine-tuning	Fine-tuning	Freezing	Freezing	Freezing	Freezing	Fine-tuning
FC1	Modification	Modification	Modification	Modification	Modification	Modification	Modification	Modification	Modification	Modification
FC2	Modification	Modification	Modification	Modification	Modification	Modification	Modification	Modification	Modification	Modification
FC3	Modification	Modification	Modification	Modification	Modification	Modification	Modification	Modification	Modification	Modification

**Table 4 sensors-25-01645-t004:** Explanation of transfer learning parameter settings.

Operation	Description
Freezing	Set the WeightLearnRateFactor parameter to 0 for the specified layer in the neural network.
Fine-tuning	Set the WeightLearnRateFactor parameter to 1 for the specified layer in the neural network.
Modification	Set the OutputSize parameter to the number of output classes for the specified layer in the neural network.

**Table 5 sensors-25-01645-t005:** Detection results of ten transfer learning methods.

Transfer learning Methods	TL1	TL2	TL3	TL4	TL5
CR (%)	64.89	92.00	91.56	99.11	97.78
Training time (s) (225 images) Training time/image (s)	1577.487 (7.01)	1699.646 (7.55)	1785.651 (7.94)	1789.566 (7.95)	1918.132 (8.53)
Transfer learning Method	TL6	TL7	TL8	TL9	TL10
CR (%)	55.56	53.78	88.44	76.89	96.44
Training time (s) (225 images) Training time/image (s)	1818.749 (8.08)	1867.853 (8.30)	1843.002 (8.19)	1913.706 (8.51)	2045.057 (9.09)

**Table 6 sensors-25-01645-t006:** Parameter settings for the detection model in this study.

Model Parameter	Setting Value	Model Parameter	Setting Value
Optimizer	SGDM	Training batch size	10
Learning rate	0.001	Training epochs	20

**Table 7 sensors-25-01645-t007:** Comparison table of large sample size and reduced sample size of the new product in the two datasets.

Domain Dataset (Product)	Source Domain Dataset (Ratchet Wrench)		Target Domain Dataset (Flex-Head Ratchet Wrench)	
Training model	Pre-trained model		Target domain model	
Anomaly kinds	4		3	
Defect types	28		30	
Domain ratio (Source/Target)	1/5	1/1	1/5	1/1
Training images per type	10	30	50	30
Testing images per type	5	15	25	15
Total training images	280	840	1500	900
Total testing images	140	420	750	450

**Table 8 sensors-25-01645-t008:** Summary table of effectiveness metrics for different network models at each assembly station.

	Models	R-CNN with Initialized Weights	AlexNet + TL	VGG16 + TL	YOLO X + TL
Stations/Metrics					
First	Recall (%)	92.63	100	100	98.08
	Precision (%)	84.21	98.57	99.05	97.14
	F1-score (%)	88.22	99.28	99.52	97.61
	CR (%)	85.27	98.66	99.11	97.33
Second	Recall (%)	89.33	99.33	100	98.00
	Precision (%)	89.33	99.33	99.33	98.00
	F1-score (%)	89.33	99.33	99.67	98.00
	CR (%)	90.30	99.39	99.39	98.18
Third	Recall (%)	68.00	72.00	73.33	68.57
	Precision (%)	85.00	90.00	91.67	87.27
	F1-score (%)	75.56	80.00	81.48	76.80
	CR (%)	88.00	92.00	93.33	90.00

**Table 9 sensors-25-01645-t009:** Summary table of training and testing time for different network models at each assembly station.

	Models	R-CNN with Initialized Weights	AlexNet + TL	VGG16 + TL	YOLO X + TL
Stations/Time					
First	Training time (s) (450 images)	2190.484	1518.006	6082.631	4084.945
	Testing time (s/image)	4.737	4.749	31.860	0.1716
Second	Training time (s) (360 images)	2461.461	1846.694	7283.576	3480.782
	Testing time (s/image)	3.906	3.856	29.883	0.1929
Third	Training time (s) (150 images)	509.127	396.735	1655.375	1520.777
	Testing time (s/image)	4.400	3.890	27.121	0.1901

**Table 10 sensors-25-01645-t010:** Summary table of training images for good and defective items at each station in the target domain.

Proportion of Target Domain Images	17%		33%		50%		67%		83%	
Product types	Good	Defective	Good	Defective	Good	Defective	Good	Defective	Good	Defective
First station	10	140	20	280	30	420	40	560	50	700
Total	150		300		450		600		750	
Second station	10	110	20	220	30	330	40	440	50	550
Total	120		240		360		480		600	
Third station	10	40	20	80	30	120	40	160	50	200
Total	50		100		150		200		250	

**Table 11 sensors-25-01645-t011:** Summary table of correct classification rates for each station under various proportions of training images in the target domain.

Station (Part Similarity)	Target Domain 83%	Target Domain 67%	Target Domain 50%	Target Domain 33%	Target Domain 17%	Average CR (%)
First station CR (%) (50% similarity)	98.67	98.33	98.66	88.67	85.33	93.93
Second station CR (%) (60% similarity)	99.27	99.50	99.39	96.36	90.91	97.09
Third station CR (%) (17% similarity)	97.60	94.74	92.00	82.61	80.00	89.39

## Data Availability

The data will be made available on request.

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
