# Peer review of "A Deep Transfer Learning-Based Visual Inspection System for Assembly Defects in Similar Types of Manual Tool Products"

_sensors, 2025, doi:10.3390/s25061645_

Round 1
Reviewer 1 Report
Comments and Suggestions for Authors
Thank you for the opportunity to review the paper.
The paper presented a study on an inspection system for manual tool assembly.
The paper is well-written. The results are also supported with good experimental methodologies and evaluations. There are also good comparisons between proposed system and other models. The reviewer has a few comments and questions as below:
Although the paper mentioned that it is to address the challenges of manual assembly of tools by suggesting automated defect detection system, it focuses mainly on the machine vision for defect detection system excluding the automation or robotic portion. The integration of the paper with robotic could be future work to consider.
The use of fixture mentioned in section 3.1, although it is clear that it aids in the placement of the test workpiece, but will it also cover certain part of the workpiece. these parts of the workpiece are not crucial and is excluded from the inspection?
In this paper, noted on the various discussion on the correct classification rates. How about a discussion on the various evaluations such as F1 score on precision, recall and confusion matrix.
There is also a discussion on the effectiveness by halfing the sample sizes. Speed is important in the automation processes. Can the authors also discuss on the comparison of time taken for training and also time taken for classification for each methods?
Reviewer 2 Report
Comments and Suggestions for Authors
This paper presents a deep migration learning based visual inspection system for detecting assembly defects in similar types of hand tool products. The study employs the R-CNN and AlexNet architectures, combined with migration learning, to show how pre-trained models can be applied to defect detection of new products with a small number of new samples and fine-tuning. The experimental results show that the AlexNet model achieves a classification accuracy of 98.67% at multiple assembly sites, outperforming the R-CNN model with randomly initialised weights. The research is innovative and practical, especially in the field of industrial automated inspection. However, the article has some deficiencies in method description, experimental design and result analysis, which need to be further improved:
1. There are inaccuracies in the language expression in the article, and some sentences are complicated in structure, which are easy to cause ambiguity. For example, in "Using transfer learning, the model can fine-tune its weight parameters with a small amount of data, making it better suited for the task in this study." In the sentence "making it better suited for the task in this study", the part of "making it better suited for the task in this study" is ambiguous, and it is not clear whether it means that the model is better suited for the task. It is not clear whether "modelling" or "fine-tuning with a small amount of data" is better suited for the task in this study.
2. Inconsistent paragraph formatting detracts from the overall readability of the article. For example, the spacing between paragraphs on pages 2 and 3 is inconsistent, with some paragraphs having a blank line and others not. It is recommended to unify the paragraph format and maintain consistent paragraph spacing to improve the overall readability of the article.
3. The diagrams (e.g., network architecture diagrams, experimental results diagrams) are not clear enough and lack the necessary labelling and explanations. For example, in Figure 7, although the network architecture diagram of the R-CNN detection system is shown, there is a lack of detailed labelling and descriptions for each part of the diagram, such as "Selective Search", "CNN Feature Extractor", "Sensor", "Sensor", "Sensor", and "Sensor". However, the diagram lacks detailed labels and descriptions for each part, such as "Selective Search", "CNN Feature Extractor", "SVM Classifier" and other parts are not clearly labelled. It is suggested to add the corresponding labels and descriptions in the figure so that readers can better understand the network architecture.
4. The derivation and explanation of formulas are not detailed enough, especially in the part of bounding box regression of R-CNN. For example, when introducing the bounding box regression of R-CNN, the article only mentions "Bounding box regression is a technique used to correct and fine-tune the position of bounding boxes. "But it does not explain the specific formula and calculation process of bounding box regression in detail. It is suggested to supplement the formula derivation and detailed explanation of bounding box regression, such as introducing how to calculate the positional offset between predicted and real boxes by linear regression model, and giving the specific formula and calculation steps.
5. The literature review focuses on surface defect detection and assembly defect detection, and lacks discussion on the application of transfer learning in industrial inspection.
6. The application of Transformer-based inspection methods in industry in recent years is not mentioned.
7. The detailed architecture diagram and process description of the combination of R-CNN and AlexNet are missing in the method description.
8. The specific implementation steps of migration learning are not detailed enough, and the description of the number of frozen layers and fine-tuning parameter settings are missing. Please provide an in-depth analysis of the performance change of migration learning under different similarities.
9. The analysis of the experimental results is rather superficial, and the reasons for the improvement of the model performance are not explored in depth.
10. The reason why AlexNet can still maintain a high accuracy rate after reducing the sample size is not explained in detail.
The language is relatively fluent, but there are a few inaccuracies in expression
